# Attosecond spectroscopy reveals alignment dependent core-hole dynamics in the ICl molecule

Hugo J. B. Marroux [1,2,6 ✉], Ashley P. Fidler[1,2], Aryya Ghosh[3], Yuki Kobayashi [1], Kirill Gokhberg[3], Alexander I. Kuleff [3,4], Stephen R. Leone [1,2,5 ✉] & Daniel M. Neumark [1,2 ✉]

The removal of electrons located in the core shells of molecules creates transient states that live between a few femtoseconds to attoseconds. Owing to these short lifetimes, time-resolved studies of these states are challenging and complex molecular dynamics driven solely by electronic correlation are difficult to observe. Here, we obtain few-femtosecond core-excited state lifetimes of iodine monochloride by using attosecond transient absorption on iodine $4d^{-1}6p$ transitions around 55 eV. Core-level ligand field splitting allows direct access of excited states aligned along and perpendicular to the ICl molecular axis. Lifetimes of $3.5 \pm 0.4$ fs and $4.3 \pm 0.4$ fs are obtained for core-hole states parallel to the bond and $6.5 \pm 0.6$ fs and $6.9 \pm 0.6$ fs for perpendicular states, while nuclear motion is essentially frozen on this timescale. Theory shows that the dramatic decrease of lifetime for core-vacancies parallel to the covalent bond is a manifestation of non-local interactions with the neighboring Cl atom of ICl.

[1] Department of Chemistry, University of California, Berkeley, CA 94720, USA. [2] Chemical Sciences Division, Lawrence Berkeley National Laboratory, Berkeley, CA 94720, USA. [3] Theoretische Chemie, PCI, Universität Heidelberg, Im Neuenheimer Feld 229, 69120 Heidelberg, Germany. [4] ELI-ALPS, W. Sandner utca 3, Szeged 6728, Hungary. [5] Department of Physics, University of California, Berkeley, CA 94720, USA. [6] Present address: Laboratoire de Spectroscopie Ultrarapide (LSU) and Lausanne Centre for Ultrafast Science (LACUS), Ecole Polytechnique Fédérale de Lausanne, ISIC, FSB, Station 6, CH-1015 Lausanne, Switzerland. ✉email: hugo.marroux@epfl.ch; srl@berkeley.edu; dneumark@berkeley.edu

High-energy photons in the soft x-ray regime access elec-trons located in core-shells of atoms and molecules. In atoms, excited states resulting from the excitation of a core-shell electron to an outer atomic orbital (core-excited state) decay in tens of femtoseconds or less[1,2]. The decay mechanism, known as Auger decay, is a pure electronic process driven by electronic correlation and consists of the filling of the core-vacancy by an outer electron while a secondary electron is emitted in order to conserve energy. The physics of Auger decay in most atomic systems is well established but in molecular systems, complicated symmetry and nuclear effects are anticipated that have so far eluded detailed experimental observation[3].

Measurement of these excited state lifetimes in molecules through linewidth studies is challenging as lineshape analysis requires consideration of unresolved vibrational structure in addition to lifetime broadening[4]. Attosecond spectroscopy offers the possibility to measure pure electronic dynamics in atoms[5] and molecules[6] with unprecedented time resolution. With this time resolution, one can directly measure core-excited-state lifetimes in the time domain[7] but thus far only isolated atoms[8,9] or strong field-related effects in core-excited molecular systems have been investigated[10]. Here, we apply attosecond transient absorption spectroscopy (ATAS) to investigate the decay of core-excited states in the ICl molecule. We find that electronic decay occurs before significant nuclear motion, and that the alignment of the core-level orbital with respect to the internuclear axis has a large effect on lifetimes. These results, observable only by time domain techniques, demonstrate that the decay of molecular core-excited states can be purely electronic processes with timescales strongly affected by molecular structure.

In heteronuclear molecular systems, extreme ultraviolet (XUV) or soft x-ray absorption at an element edge creates a vacancy in a specific atom. The decay mechanisms of the localized core-hole can be divided into local and non-local channels[11]. In the local channels, all the electrons involved originate from the atom bearing the vacancy and these channels are analogous to Auger decay in isolated atoms. In non-local channels, the secondary electron emitted is either localized on another atom or delocalized over multiple atoms. Non-local channels occur via processes that are analogous to interatomic Coulombic decay[12] or electron-transfer-mediated decay[13] mechanisms in weakly bound systems, and they can represent a significative portion of the decay process as they often lead to lower energy cations.

The few time-resolved experiments on non-local decay chan-nels obtain lifetimes >100 fs[14–16], but those were focused on the study of inner-shell transitions of rare-gas dimers. In covalently bound molecules, the small distances between the atom in which the initial excitation is created and the neighboring atoms can bring the lifetime of the core-hole down to the few or sub-femtosecond regime[4]. The resulting large lifetime broadening of the absorption lineshape combined with the complex lineshape of electronic transition in molecules (vibronic progressions, sym-metry, etc.) prevent the direct use of frequency-domain techni-ques to measure molecular core-excited stated lifetimes[4].

Here, we employ a time-domain approach using ATAS to investigate iodine $6p \leftarrow 4d$ core-to-Rydberg transitions around 55 eV in ICl. Energetically distinct states with iodine core-hole orbitals aligned parallel or perpendicular to the molecular axis are measured[17,18]. Core-hole orbitals aligned parallel to the inter-nuclear axis exhibit significantly shorter lifetimes (3.5 ± 0.4 and 4.3 ± 0.4 fs) compared to core-hole orbitals perpendicular to the molecular axis (6.5 ± 0.6 and 6.9 ± 0.6 fs). During the timescale of these decays, the bond length changes by at most 3.5% (0.075 Å), so nuclear motion is minimal. The lifetime dependence on core-hole orbital alignment is reproduced in part by ab initio calcu-lations using the Fano-algebraic diagrammatic construction

(ADC)—Stieltjes method[19,20]. Results from the calculation attri-bute the differences in decay rates to a greater participation of delocalized molecular orbitals (MO) (non-local effect) in the decay of core-excited states aligned along the molecular axis. The observation of excited state decays that are faster than nuclear motion and the dependence of decay rates on orbital alignment opens an uncharted field of investigation exploring electronic molecular decay dynamics using attosecond spectroscopy.

## Results

**Static measurement.** The absorption spectrum of ICl corre-sponding to $4d^{-1}6p$ Rydberg excitation on the I atom is collected by spectrally analyzing an isolated attosecond pulse (IAP) transmitted through the sample as shown in Fig. 1a and is reported in Fig. 1b. This spectrum is similar to published spectra of halogen-containing diatomic molecules[17,21]. As shown in Fig. 1c and deduced from published photoelectron spectrum[22], the iodine $4d$ core-levels are split by 1.7 eV due to spin–orbit coupling, and each spin–orbit level has a ligand-field splitting of 0.3 eV[22]. The four peaks in Fig. 1b are labeled by $\Omega_c$, the pro-jection of the core hole orbital and spin angular momentum along the internuclear axis[23]. The core-hole orbitals in Fig. 1c are dis-cussed in more detail below. The spectral resolution is better than 50 meV and does not contribute significantly to the lineshapes.

**Time-resolved measurement.** In ATAS, an IAP covering the iodine $N_{4,5}$ edge of ICl is linearly absorbed by the sample and dispersed on a spectrometer as shown on Fig. 1a. Identical to the optical domain, this absorption step creates a macroscopic polarization in the sample that decays with a dephasing time that, in the gas phase, is directly linked to the lifetime of the core-excited state[24]. The decay of the polarization, termed free induction decay (FID), results in an absorption feature in the IAP. In the time domain, the FID is longer lived than the IAP and can be perturbed by a delayed NIR pulse with sub-4 fs duration. The perturbed FID induces changes in the absorption lineshapes[25] that are isolated using a shutter to record the IAP spectra with and without the NIR pulse. The transient absorption is then computed using $\Delta A = -\log_{10}(I_{on}/I_{off})$, reported here in $\Delta$mOD (changes in milli-optical density).

By scanning the delay between the IAP and the NIR pulse, core-hole lifetimes can be retrieved from the polarization decay, as has been shown for inner valence excited states of argon and xenon[26,27] as well as core states of krypton[9]. ATAS was recently applied to core-excited states of methyl iodide, where transitions to Rydberg excited states dominate the spectrum, but no core-hole lifetimes were reported[10].

Figure 1d shows the ATAS spectrum of ICl for various time delays acquired using the experiment described previously[28] and in the "Methods" section. Here, positive time delays correspond to the XUV pulse arriving before the NIR pulse. The signal is composed of negative transient features at central frequencies of the transitions observed in the static spectrum (Fig. 1b), and positive features on either side of these.

Perturbation of the FID can proceed via laser-induced ionization of the decaying dipole[29], resonant coupling with a dark state[30], or non-resonant AC stark shifting of the excited state[25]. In the work reported here, single photon ionization is inaccessible with the NIR pulse and the peak power was kept low enough ($2 \times 10^{13}$ W/cm$^2$) to minimize ionization by strong field processes. The $6s$ state, which is the principal candidate for resonant coupling with the $6p$ excited states, is too close in energy (0.8 eV) and cannot be populated by the broadband NIR photon energy[17]. In a study of methyl iodide, Drescher et al.[10] made

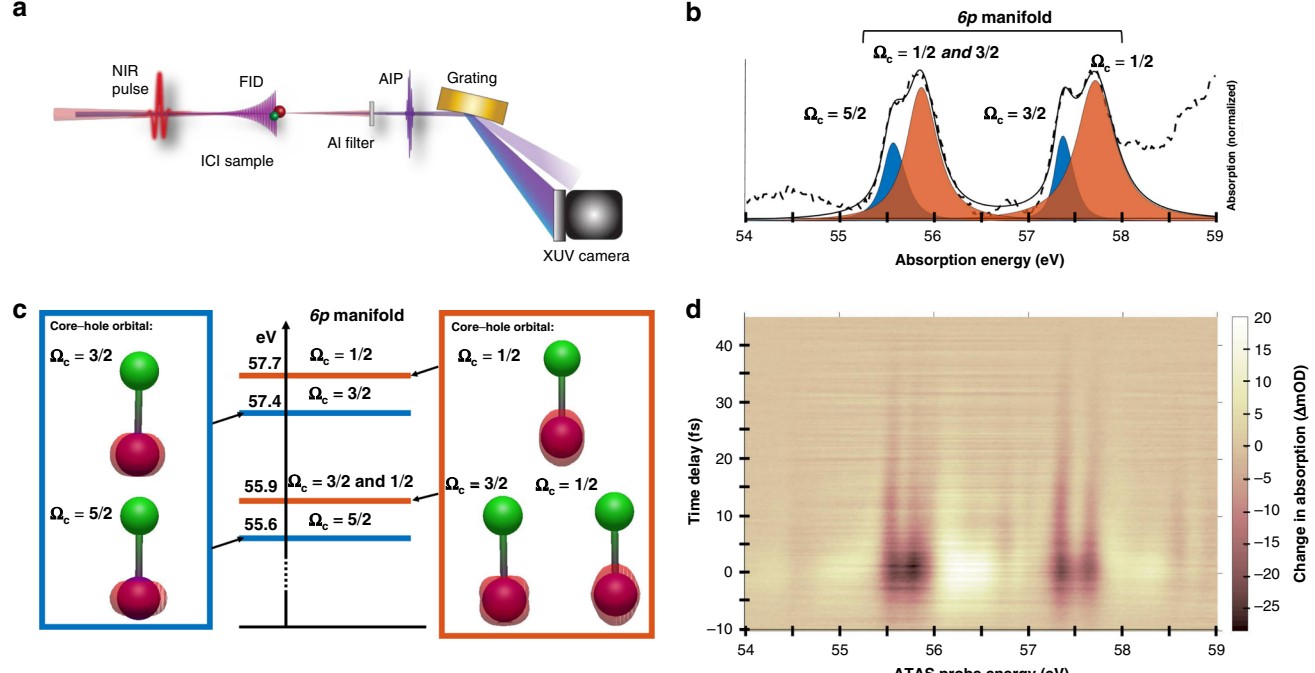

**Fig. 1 Attosecond transient absorption of ICl molecules. a** Key components of the ATAS experiments. The interaction of the IAP with the sample creates the free induction decay (FID) shown in purple. Delayed NIR pulses perturb the FID, which affects the spectrum of the IAP recorded on the XUV camera and interferes with the incident pulse. The NIR pulse is removed after the sample using an aluminum filter (cf. text for details). Changes in absorption are isolated using an optical shutter to modulate the NIR pulse (not shown). **b** Iodine $N_{4,5}$ edge absorption of ICl centered around $4d^{-1}6p$ transitions obtained with the described experiment (dashed line). The orange and blue features are the fits to transitions to the indicated $4d^{-1}6p$ core-excited states. The lineshapes include lifetime broadening, finite spectral resolution and vibronic progressions as described in the text. The solid black line is the total fit. **c** Core-hole orbital electronic density (cf. SM3 for computation details), separated into core-orbitals perpendicular to the molecular (blue box, blue features in **b**), and orbitals aligned along the molecular axis (orange box, orange features in **b**). The energies are determined based on the fittings to the experimental absorption spectrum. **d** ATAS spectrum of ICl at the iodine $N_{4,5}$ edge in ΔmOD (changes in milli-optical density, cf. text for details).

similar observations and concluded that a non-resonant (i.e. Stark shift) interaction is responsible for the transient signal.

In order to confirm that non-resonant coupling of the NIR pulse is the origin of the transient signal, the time-zero transient spectrum has been simulated in Fig. 2a by considering that the NIR pulse induces a shift in the phase (Δφ) of the FID that is proportional to the ponderomotive energy of the NIR field: $\Delta\varphi(t, \tau) = \int \frac{[E_0(\tau,t')]^2}{4\omega^2} dt'$, where $E_0$ and $\omega$ are the NIR field amplitude and central frequency, respectively[25,30,31]. The time-dependent Schrödinger equation is solved considering only the four main resolvable transitions from the core-levels to the $6p$ Rydberg state. Here the features are uniformly lifetime-broadened to match the experimental spectrum and no vibronic or experimental broadening is taken into account. The main features of the experimental spectrum (in red) are reproduced in the simulation (in gray) confirming that the non-resonant interaction dominates.

The exponential recovery of the negative depletion features depends on the dephasing time of the FID of the considered transition[27]. In the gas phase, dephasing is limited by the population lifetime ($T_1$) of the excited states so that the FID mentioned earlier is $p(t) \propto \exp(-t/2T_1)$[27]. Hence, the FID decay occurs at half the rate of the population decay. For simplicity, the factor of two between population and dephasing is included in all the time constants reported here and only population lifetimes are discussed (dephasing time constants are reported in Supplementary Table 3 for completeness). Different experimental parameters and checks required to ensure that accurate lifetimes are measured are discussed in the "Methods" section. The kinetic traces yielding lifetimes are obtained by taking lineouts at each of

the central frequencies of the absorption features and fitted to an exponential recovery function convoluted with a Gaussian to capture the finite time resolution of the experiment.

The central energies of the four transitions discussed here are determined by fitting the static spectrum with lineshapes obtained by considering expected vibronic progressions, convoluted with the 50 meV spectral resolution and varying the values of the line broadening due to lifetime (cf. SM1 for details). Figure 2b, c then show the kinetic fits at each feature's central frequencies reported at the positions of the arrows in Fig. 2a. In ICl, the decomposition of the static spectrum in Fig. 1b indicates that the features partially overlap, potentially leading to errors in measured lifetimes. This analysis is considered in SM2.

**Spectral assignment**. The states investigated are located near the iodine $N_{4,5}$ edge and correspond to $4d^{-1}6p$ core-excited states. The spectral assignment is performed using ab-initio calculations including the spin–orbit coupling induced by the iodine atom. In this context and similarly to other halogen containing diatomic molecules[21], $4d$ core-levels of iodine are split into five different core-levels shown in Fig. 1c by a combination of ligand field and spin–orbit splitting. The composition of the resulting core holes in terms of the atomic orbitals ($d_{xz}$, $d_{yz}$, $d_{z^2}$, etc.) is reported in SM3 and is similar to results from previous calculations[22]. This decomposition shows that the main components of the $\Omega_c = 5/2$, $3/2$ orbitals in the blue box are the $d_{x^2-y^2}$ and $d_{xy}$ orbitals, where the z-axis is located along the internuclear axis. The high values of the angular momentum projection ($\Lambda = 2$) of these orbitals on the internuclear axis indicate that the electron density of these two orbitals is aligned perpendicular to the molecular axis. On the

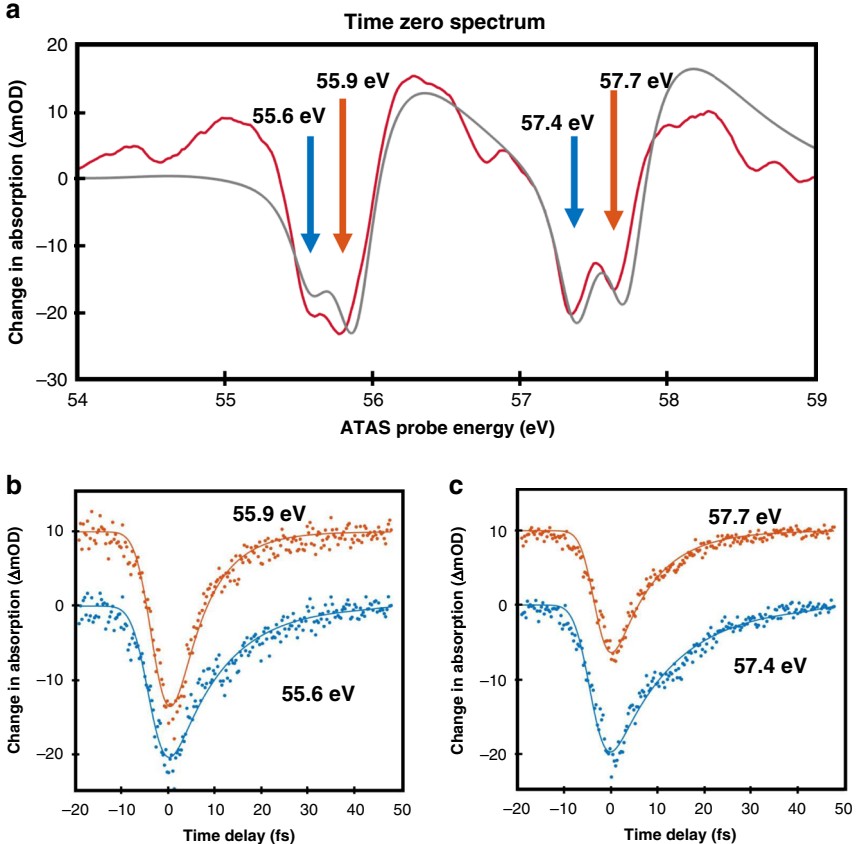

**Fig. 2 ATAS spectra simulation and kinetics. a** Time zero ATAS experimental (in red) and simulated (in gray) spectra of ICl. The arrows show the positions of the resonances obtained from the fit of the spectrum in Fig. 1b. Kinetic traces of the depletion recovery at the resonance energy are shown as dots in **b**, **c**. As described in the text, the kinetic traces at 55.9 and 57.7 eV can directly be linked to the lifetimes of core holes. Traces obtained at 55.6 and 57.4 eV have overlapping contributions as described in the main text and in SM2. The orange traces in **b**, **c** have been displaced vertically by 10 ΔmOD for clarity. Solid lines are fits obtained using the convolution of an exponential recovery and gaussian function.

other hand, the three $\Omega_c = 3/2$, 1/2 orbitals in the orange box are mainly composed of the $d_{xz}$, $d_{yz}$ and $d_{z^2}$ orbitals, which have angular momentum projection of 1 and 0 indicating an electron density more parallel to the molecular axis.

Experimentally, due to the lineshape broadening of ~130 meV from vibrational effects, and a further 75–180 meV of lifetime broadening (depending on the states), only two types of core-hole alignment are discernible. The two core-orbitals with $\Omega_c = 3/2$, 1/2 in the lower part of the orange box in Fig. 1c and located at 55.9 eV are nearly degenerate and are treated together. The three other orbitals can be spectrally isolated and are treated independently.

Nominally, $6p$ Rydberg orbitals are assigned to $6p\sigma$ and $6p\pi$ states. Due to the large radius of Rydberg orbitals, the $6p\sigma/6p\pi$ splitting is too small (<50 meV) compared to the transition linewidths to be resolved. Both orbitals will thus be referred to as $6p$ without further distinction. Transitions to other Rydberg states, e.g. $6s$ or $7p$, are possible, but because of the relatively weak transition dipole moments to these states, transitions to $6p$ Rydberg states dominate the static absorption spectrum in the spectral region considered[17].

**Molecular dynamics**. From the measured lifetimes, the timescales for electronic decay are much shorter than nuclear motion. The core-excited states discussed here are bound and the computed I-Cl stretch frequencies are between 420 and 430 cm$^{-1}$ depending on the state considered (cf. SM1 for details). Hence, the half-vibrational period is ~40 fs and the nuclear displacement for a half-vibration is 0.34 Å, as inferred from the

potential energy curves shown in Supplementary Fig. 1a. Depending on the state considered, the timescale for the population lifetime gives a variation on the average internuclear distance (2.32 Å) of between 1% and 3.5% during one time constant of the electronic decay. This leads to the conclusion that the core-hole decays in ICl are an example of nearly pure electronic molecular dynamics.

Measured lifetimes show substantial dependence on the alignment of the core-hole MO relative to the molecular axis. As shown in Table 1, core-excited states aligned parallel to the covalent bond, i.e. $\Omega_c = 3/2$, 1/2 in orange in Fig. 1c, are 1.6 and 1.9 times shorter-lived than the states aligned perpendicular to the molecular bond ($\Omega_c = 3/2$, 5/2 in the blue box). A similar effect was computed for the decay of van der Waals dimers[32,33] and points to a manifestation of a non-local effect on the iodine core-hole states. To confirm the presence of these effects in ICl we consider the decay channels open to the core-excited states.

The $4d^{-1}6p$ excited state can decay by two types of pathways as represented in Fig. 3: participator channels (channel i in Fig. 3), where the electron in the $6p$ Rydberg orbital is ionized or moves to a lower-lying orbital, and spectator channels (channels ii and iii in Fig. 3), where only the valence and core-electrons are involved in the decay. Even though participator channels lead to lower energy products, they are known to be minor in atomic iodine[34] and xenon[35]. Spectator channels are favored in these atoms because of the stronger Coulombic interaction of the core-electron with valence electrons relative to the electron in the

**Table 1 Iodine $4d^{-1}6p$ core-excited staes lifetimes in ICl.**

| Hole orientation | Perpendicular | Parallel | Perpendicular | Parallel |
|---|---|---|---|---|
| Core-hole character | $\Omega_c = 5/2$ | $\Omega_c = 3/2, 1/2$ | $\Omega_c = 3/2$ | $\Omega_c = 1/2$ |
| Energy (eV) | 55.6 | 55.9 | 57.4 | 57.7 |
| Lifetime (fs) from ATAS measurement | 6.5 ± 0.6 | 3.5 ± 0.4 | 6.9 ± 0.6 | 4.3 ± 0.4 |

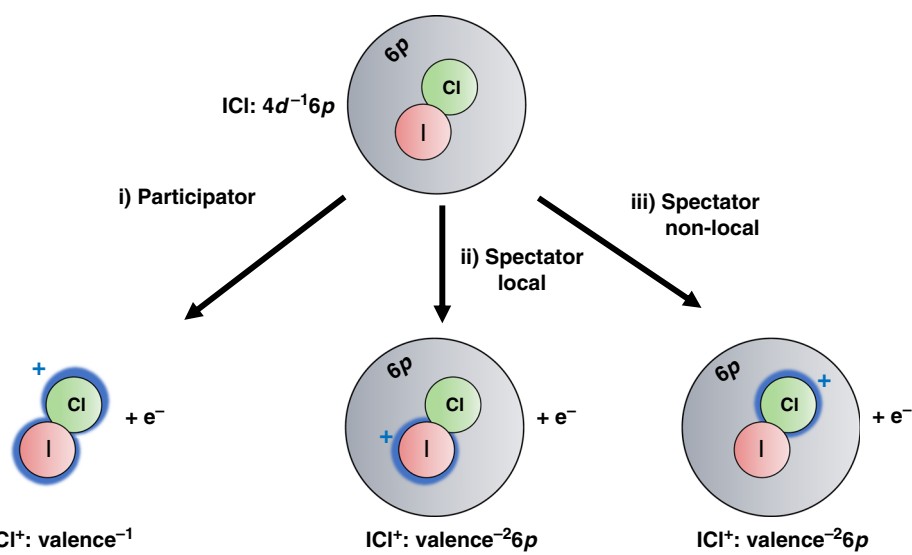

**Fig. 3 Open decay channels for $4d^{-1}6p$ core-excited state.** Channel i represents the participator channels where the electron in the Rydberg $6p$ orbital (represented in gray) is involved in the electronic decay. The local and non-local channels of the spectator decay, channels ii and iii respectively, lead to a partial charge on the iodine or chlorine atoms, as illustrated by the blue halos.

delocalized $6p$ Rydberg state. The situation in ICl is similar so the participator channels are not expected to contribute significantly to the core-excited state decay.

To understand the variation of lifetime with core-hole alignment, ab initio calculations of the core-hole partial linewidth were conducted using the Fano-ADC-Stieltjes method as detailed in the "Methods" section[19,20]. The calculation yields the partial linewidth associated with each open decay channel, which can be converted to lifetime using $\tau = \hbar/\Gamma$, where $\Gamma$ is the linewidth, and $\tau$ the lifetime. The method relies on a non-relativistic Hamiltonian, so effects such as spin–orbit interaction cannot be reproduced. Decomposition of the orbitals shown in Fig. 1c into $d$-orbitals excluding spin–orbit interactions are shown in SM3.

The calculation of partial linewidths of core-excited states is difficult due to the complexity of the final state manifold. However, as core-hole relaxation is expected to be dominated by spectator decay, it is a reasonable approximation to consider the core ionized molecule rather than the $4d^{-1}6p$ state of the neutral. Therefore, we computed partial linewidths of ICl cations with a hole in the $4d$ orbital. In these conditions, the three cationic states considered are the $^2\Sigma$, $^2\Pi$, and $^2\Delta$ states, which correspond to the ionization limits of the $4d\sigma np$, $4d\pi np$, and $4d\delta np$ Rydberg series, respectively.

At the energy considered, 48 channels are open (cf. SM4). The decay channel showing the largest variation of partial linewidth with the type of initial core-hole is displayed in Fig. 4a. Given that the calculation considers core-excited cations as initial states, decay products are doubly charged with the two holes located in available valence MOs. For the channel considered in Fig. 4a, the two holes of this decay product are located in the same valence

MO shown in the inset of Fig. 4a. For this final state, partial linewidths for the $^2\Sigma$, $^2\Pi$, and $^2\Delta$ core-holes are 14.1, 5.3, and 2.3 meV, respectively. The evolution of the linewidths with the core-hole types shows the decrease of the contribution of this final state to the core-hole decay. The main atomic orbital contributing to the empty MO shown is the Cl $3p$ (80% of the total orbital) so the contribution of non-local effects on the iodine core-hole decay is major for this channel. This highlights the role of non-local effects and core-orbital alignment on the core-hole decay.

In order to compare experimental and computational results, all the decay channels have been considered and the MO of all the product and partial widths are shown in SM4. The total linewidths of each of the hole types are obtained by summing the partial linewidths of all open decay channels.

The lifetimes computed from the total widths are shown in Fig. 4b and are approximately twice as short as the experimental ones. This is due to a known bias in the computational method and consistent with other known cases[36–38]. Despite the approximations in the calculations (no spin–orbit interaction, considering the molecular cation, etc.), the relative simulated lifetimes capture the dependence on the core-hole alignment at various internuclear distances. At the equilibrium distance, $(R_{eq} = 2.32 \text{ Å})$[39], the computed lifetime of the $^2\Sigma$ state is 1.25 times shorter than the lifetimes of $^2\Pi$ and $^2\Delta$. This can be compared to the ratio of the $\Omega_c = 3/2, 1/2$ core holes lifetime at 55.9 eV and the $\Omega_c = 5/2$ core-hole lifetime at 55.6 eV measured by ATAS of 1.9. The lifetimes computed for the cations in the $^2\Pi$ and $^2\Delta$ states at the equilibrium bond distance are similar despite the differences in the core-hole wavefunction alignments. This may reflect inaccuracies in the lifetime calculation due to the

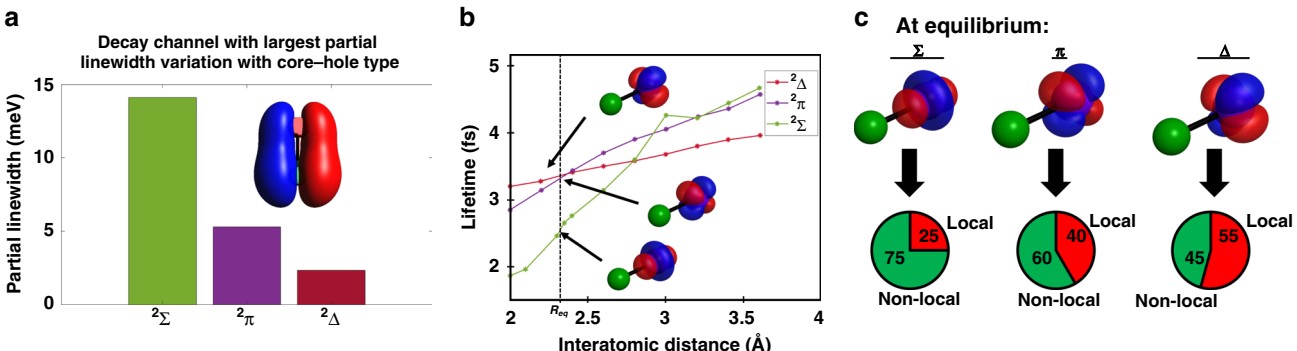

**Fig. 4 Origin of the lifetime variation. a** Partial linewidths of each of the core-hole types for the decay channel showing the largest partial linewidth variation. For this channel, the two valence holes of the decay product are located in the same MO shown in the inset. **b** Calculated core-hole lifetime of the cation for the three types of core-hole orbitals at various internuclear distances. Conversion between linewidth and lifetime is obtain using $\tau = \hbar/\Gamma$, where $\Gamma$ is the linewidth, and $\tau$ the lifetime. The equilibrium distance (2.32 Å) is shown by the dotted line. **c** Contribution of the local and non-local decay chanels at the equilibrium internuclear distance.

neglect of spin–orbit effects or consideration of the core-excited cation rather than the neutral species.

Computations consider the effect of the neighboring chlorine atom on the measured iodine core-hole lifetimes. In order to classify the valence delocalized orbitals into local and non-local channels, the final state empty MOs bearing the charges are projected onto atomic orbitals. Weights of the two types of channels are shown for each core-hole type at the equilibrium distance in Fig. 4c. While the $^2\Pi$ and $^2\Delta$ core-orbitals show similar contributions of local and non-local channels, the $^2\Sigma$ core-hole shows a major contribution of non-local effects.

## Discussion
Molecular core-excited states dynamics of iodine monochloride (ICl) following $6p \leftarrow 4d$ core-to-Rydberg excitation are measured using ATAS. We show that this technique gives a direct access to lifetimes of the core-hole and allows one to follow the molecular decay in real time. Four molecular core-excited state lifetimes between 3.5 and 6.9 fs are reported depending on the core-hole character.

During the decay of the core-excited states with $4d^{-1}6p$ character the nuclei move by <3.5% of their internuclear distance. Following on the ever-growing interest in charge migration[40,41] and photoionization time delays[42], this investigation shows not only that the molecular dynamics of core states can be investigated by attosecond spectroscopy, but that their decays can be a nearly pure electronic decay process.

In this regime of molecular dynamics, core-excited state lifetimes depend on the core-hole orbital alignment with respect to the molecular axis. Core-hole orbitals with $\Omega_c = 3/2$ and $1/2$ and parallel to the molecular axis exhibit shorter lifetimes than orbitals with $\Omega_c = 5/2$ and $3/2$ and perpendicular to the molecular axis. Calculations confirm this effect and attribute it to the larger contribution of non-local effects on the decay due to the presence of the nearby chlorine atoms for hole orbitals aligned along the molecular axis.

Molecular core-levels are often considered to be little influenced by the valence structure or molecular environments and are often replaced by pseudopotentials in electronic structure calculations in order to simplify relativistic effects[43]. Previous studies showed that core-level spectra can be non-degenerate due to ligand-field splitting[22]. This time-resolved study shows that their lifetimes are also greatly influenced by the molecular structure. This opens questions on the effect of lower molecular symmetry, ligand electronegativity, solvation environment, etc. on core-

excited state lifetimes. These topics relate to a wide range of fields such as chemistry and can now be addressed using attosecond spectroscopy.

## Methods

**ATAS experiment**. The laser setup was reported elsewhere[30]. Briefly, a carrier envelope phase stable Ti:Sapphire oscillator (Femtolaser, Rainbow) and multi-pass amplifier are used to produce carrier envelope stabilized pulses (1.8 mJ, 1 kHz, 25 fs at 780 nm). The laser pulses are spectrally broadened in a stretched hollow-core fiber of 2 m length and a 500 μm inner diameter (few-cycle Inc.) filled with Neon gas. Pulses are compressed using seven pairs of double angled chirped mirrors (Ultrafast Innovations, PC70) and a 3 mm-thick ammonium dihydrogen phosphate crystal to correct for third-order dispersion[44]. The compressed beam is separated by a 70/30 broadband beam splitter (LAYER-TEC) and directed toward the high-order harmonic generation (HHG) cell and probe beam, respectively. IAPs are generated by focusing the 3.7 fs long pulse (1.4 optical cycle) into a gas cell with flowing argon using a $f = 500$ mm concave mirror[44]. The driving field characterization was done using a commercially available dispersion scan, i.e. d-scan (Sphere Ultrafast Photonics) and the results are shown in SM5. The isolated character of the attosecond pulse was established by confirming that the spectrum was continuous and showed a strong variation with the laser carrier envelope phase of the driving pulse, based on previous streaking measurements[44].

The driving NIR field for the HHG is separated from the IAPs using a 200 nm-thick aluminum filter supported on a mesh (Lebow). The IAP pulses are refocused toward the sample gas cell using a gold-coated toroidal mirror. The delayed NIR pulse is recombined with the IAP between the toroidal mirror and the target cell using an annular mirror. The delayed NIR pulse is focused using a silver-coated $f = 1000$ mm concave mirror. After the target cell, the NIR probe pulse is then removed using an aluminum filter similar to that used to separate the HHG. An aberration-corrected concave grating (Hitachi, part number 001-0640) is used to disperse the light onto a CCD camera (Princeton Instrument, Pixis)[45]. The experiment is conducted using the diffraction grating in second order and the spectral resolution was determined to be 50 meV at 65 eV by fitting of the core-level transitions in xenon.

The ICl sample was purchased from Sigma Aldrich and used without further purification. Adequate sample density was achieved by heating the sample container and gas lines to 40 °C using heat tapes. Transient absorption spectra were obtained by collecting the IAP spectrum with ($I_{on}$) and without the delayed NIR beam ($I_{off}$) (modulated using a mechanical shutter) and computed using $\Delta A = -\log(\frac{I_{on}}{I_{off}})$.

**Core-hole lifetime measurement**. To reduce systematic errors in lifetime measurements due to low spectral resolution, inducing an early cancellation of the depletion feature by nearby positive features, static and time-resolved data were collected using the second order of the spectrometer diffraction grating reaching a spectral resolution of 50 meV. Power dependences and discussion of the spectral resolution are detailed in the SM6 to verify that the measurements accurately report core-hole lifetimes. To further confirm the lifetime measurement, the lifetime of core-excited xenon following excitation of its $6p \leftarrow 4d$ transition was measured using ATAS under the same laser power, pressure and spectral resolution conditions as those used in the ICl experiments. A lifetime of $5.9 \pm 0.7$ fs is measured in these conditions, in good agreement with a previous estimate from linewidth measurements, which indicated a lifetime of $6.2 \pm 0.2$ fs[46]. (cf. SM7 for spectrum and kinetic traces).

**Fano-ADC-Stieltjes method**. Ab initio calculations of the core-hole lifetime were conducted using the Fano-ADC-Stieltjes method[19,20] In the Fano-ADC-Stieltjes method, natural linewidths (Γ) are obtained by separately constructing the continuum composed of the decaying state and the leaving electron ($\chi_{\beta,\varepsilon}$), the bound initial state (Φ), and the coupling between the two. The width is given by the golden rule-like expression, where the coupling moments are summed over all open decay channels.

$$\Gamma = 2\pi \sum_{\beta} \left| \left\langle \Phi \middle| \hat{H} \middle| \chi_{\beta,\varepsilon} \right\rangle \right|^2$$

The width is then converted to lifetime following the uncertainty principle ($\tau = \hbar/\Gamma$).

As discussed in the main text, the decay of core-excited states considered mainly occurs via spectator mechanisms where the electron in the $6p$ Rydberg orbital does not participate. To simplify the calculation, core-ionized cations were, therefore, considered instead of core-excited ones. Moreover, the non-relativistic Hamiltonian has been used to construct the initial and final states and, thus, the spin–orbit interactions have been neglected. In these approximations, three core ionized states are considered: the $^2\Sigma$, $^2\Pi$, and $^2\Delta$, corresponding to the ionization limits of the $4d\sigma np$, $4d\pi np$, and $4d\delta np$ Rydberg series, respectively.

## Data availability

Data are available on request from the authors.

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

## Acknowledgements

The authors acknowledge Lorenz Cederbaum and Romain Geneaux for valuable discussions. H.J.B.M., A.P.F., D.M.N., and S.R.L. acknowledge the Director, Office of Science, Office of Basic Energy Sciences through the Atomic, Molecular, and Optical Sciences Program of the Division of Chemical Sciences, Geosciences, and Biosciences of he U.S. Department of Energy at Lawrence Berkeley National Laboratory under contract no. DE-AC02-05CH11231. A.P.F. acknowledges funding from the NSF Graduate Research Fellowship Program. Y.K., D.M.N., S.R.L., and A.I.K. acknowledge support

from the U.S. Army Research Office (ARO) (No. W911NF-14-1-0383). Y.K., and S.R.L. acknowledge support from the National Science Foundation (NSF) (CHE-1660417 and 1951317) for absorption spectra calculations Y.K. also acknowledges financial support from the Funai Overseas Scholarship. A.G., K.G., and A.I.K. acknowledge the support by the European Research Council (ERC) under the Advanced Investigator Grant No. 692657.

## Author contributions

H.J.B.M., A.P.F., D.M.N., and S.R.L. designed the experiment. H.J.B.M. and A.P.F. performed the data collection. A.G., K.G., and A.I.K. performed the core-hole lifetime calculations and Y.K. performed the absorption spectra calculations. H.J.B.M., D.M.N., and S.R.L. wrote the manuscript.

## Competing interests

The authors declare no competing interests.
