## [Peer Review File · Nature Communications]

REVIEWER COMMENTS

Reviewer #1 (Remarks to the Author):

This manuscript deals with the alignment dependent core-hole dynamics in the ICI molecule. Different lifetimes are reported after I4d resonant excitation. The experimental results are supported by theoretical simulations. Both experimental and theoretical team leaders are highly recognized scientist publishing frequently high impact publications.

For a broad interest journal like Nature Communications, the manuscripts have to be didactic. I had pain to understand. A high effort has to be made when reading the manuscript. As an illustration, the second sentence is explaining with difficulties the Auger decay. A big effort has to much better explain.

I'm not expert in this field. I had troubles to determine how important are the presented results. A better presentation of what is known so far for the lifetimes would be appreciated.

Since the Auger decay is involving 2 delocalized valence electrons, I have troubles to consider the electronic decay as atomic-like as claimed in the second paragraph of the introduction.

For me, the fourth paragraph is difficult to catch. The authors need to find a way to describe briefly the "macroscopic polarization". Without reading the reference 16, it is impossible to get an idea about this interesting concept.

Concerning the results in figure 1, it would be interesting to know the experimental resolution, How an absorption spectrum measured with Synchrotron Radiation would look like ? Some signal seem to increase at 54.5 eV. Is it background ? Is it a discrete state ? Same question for the signal at 58.5 eV. I do not know the DELTA mod scale. I did not find explanation about this scale in the manuscript. How is it related to the delay ?

In the figure 2, the absorption spectrum is now upside down. Why ? For clarity, I advise to not change the orientation from figure 1 to figure 2. Again this DELTA mod scale on the y axis.

Page 7, I did not understand how the lifetime are determined. Is it a Voigt deconvolution ? I believe the most important questions concerns the understanding of the IR transition, the spectator Auger decay and the ICD-ETMD processes. It is very difficult to realize the details of the transitions, how do they evolve with the delay ?

As a conclusion, there is a lot of work to rephrase the manuscript in order to make it understandable by a broad audience.

Reviewer #2 (Remarks to the Author):

In the work "Attosecond spectroscopy reveals alignment dependent core-hole dynamics in the ICI

molecule” by Marroux et al., the authors present the results of experiments using an isolated attosecond pulse (IAP) and a few-cycle infrared pulse to time-resolve core-excited state dynamics in iodine monochloride molecules. Through measurements based on attosecond transient absorption spectroscopy (ATAS) they directly access the core-hole lifetimes with few-femtosecond resolution, demonstrating a dependence on the core-hole character, on timescales where the positions of the nuclei are essentially frozen. The experimental results are compared to thorough theoretical investigations, supporting the conclusion that the lifetimes are strongly affected by ligand-field effects, and thus on the alignment of the hole orbitals relative to the molecular axis.

The work is, to the best of my judgement, scientifically and technically sound and the combination of experimental and theoretical results strongly support the conclusions. The approach is novel in the sense that it is the first demonstration of ATAS of a weak-field process in core-excited molecules. I consider the work to be an important contribution to scientists in the field, in that it clearly demonstrates the precision and potential of ATAS measurements, and for the wider audience it opens up for few-femtosecond studies using ATAS in systems of interest in a range of science fields.

To that end, I recommend publication of the submitted manuscript in Nature Communications, but recommend the authors to consider the comments below in order to improve the accessibility and clarity of the manuscript:

1. Reading through the manuscript text, it is crucial to the understanding to keep track of the four different channels that the study is focusing on. I think that the notation and labelling of these between the different figures and between the text and the figures can be improved. For example, in Figure 1 they are referred to by their spectroscopic notation, in Figure 2 they are numbered 1-4 (and blue/red for the alignment character), in table 1 again with spectroscopic notation but no numbers and, finally, in the figures and table of the SM, they are sometimes referred to only by their energies. I encourage the authors to think about whether the clarity of this can be improved to facilitate for the reader. Maybe something like $1/2_{\text{par/perp}}$ or similar?

2. I think that Figure 1 would benefit from some additional information. First, by adding axis labels in addition to the units. Second, Figure 1a is the same as Figure S2c (in the SM), and it would make sense, in the figure caption, to also describe the contents of Figure 1a) similarly to how it is done for Figure S2c (e.g. what is the red, blue, dashed, solid lines?).

3. I find it slightly difficult to follow the discussion around the different times involved. The fitted times (directly from the kinetic traces), the dephasing times (after subtracting the overlapping traces) and, finally, the lifetimes (after division by the factor of two due to the difference between the decay of the macroscopic polarization and the population lifetime). I think that it could be beneficial to present all of this in a single table in the SM, including all the times in a step-wise fashion, i.e. merging tables S2 and S3 (maybe even table S1 with the results from the lineshapes?) and adding the dephasing time as an intermediate step. In addition, I find that the reason for the factor of two is explained in the main text, with reference to the appropriate source, while in the SM it is (as far as I can find) only referred to as “Accounting for the factor of two...”. I think it would be useful to repeat the explanation and provide the reference again in the SM.

Reviewer #3 (Remarks to the Author):

This manuscript describes a beautiful study of core-hole decay dynamics in ICl using attosecond transient absorption spectroscopy (ATAS). The authors measure the lifetimes of $4d^{\wedge}16p$ Rydberg states of ICl and find lifetimes ranging from 3.5 to 6.9 fs. They convincingly argue that these decays are purely electronic with structural dynamics playing a negligible role.

The data is convincing, the paper is clearly and nicely written and the overall interpretation is plausible. The main issue I have is the fact that spin-orbit coupling is neglected everywhere, both in the qualitative interpretation and in the calculations. I have some doubts that this is correct and it might have implications on the interpretation.

First, the pictorial representation of the d orbitals as $d_{z^{\wedge}2}$, $d_{x^{\wedge}2-y^{\wedge}2}$, etc. completely neglects spin-orbital coupling. By analogy with p orbitals, it is clear that the spin-orbit-coupled eigenfunctions are actually linear combinations of the pure angular-momentum eigenfunctions, which are currently shown. This will change the qualitative shape of the orbitals, even in an atom. Since this is very easy to check, I suggest that the authors show the spin-orbit-coupled eigenstates, instead.

Second, and this is related, the authors use a Hund's case (a) notation throughout the paper, where a Hund's case (c) notation would be needed. Because the spin-orbit splitting is completely dominant here, λ cannot be a good quantum number, only ω can be. As a consequence, also S (which should be Σ in Hund's case (a)) will not be a good quantum number. I suggest that the authors change the quantum-number notation to be consistent with Hund's case (c), or explain their choice of the current notation.

Third, and this might well be too demanding, it would be great to see some theoretical results including spin-orbit coupling.

Finally, I would like to suggest that the authors also reference previous work done on studying purely electronic dynamics in molecules using high-harmonic spectroscopy and RABBIT.

Reviewers comments on the manuscript: Attosecond spectroscopy reveals alignment dependent core-hole dynamics in the ICl molecule.

Reviewer #1 (Remarks to the Author):

This manuscript deals with the alignment dependent core-hole dynamics in the ICl molecule. Different lifetimes are reported after I4d resonant excitation. The experimental results are supported by theoretical simulations. Both experimental and theoretical team leaders are highly recognized scientist publishing frequently high impact publications.

For a broad interest journal like Nature Communications, the manuscripts have to be didactic. I had pain to understand. A high effort has to be made when reading the manuscript.

As an illustration, the second sentence is explaining with difficulties the Auger decay. A big effort has to much better explain.

I'm not expert in this field. I had troubles to determine how important are the presented results. A better presentation of what is known so far for the lifetimes would be appreciated.

We thank the reviewer for pointing out the need for improving the didactic aspect of our manuscript. It is important to us that our work reaches beyond our native community. We have rephrased the introduction of our paper in order to better explain the different physical processes at play (Auger decay, non-local decay, core-hole lifetime etc..) and their importance. We also included a new figure to illustrate the main components of the ATAS measurement. These changes have as a goal to facilitate the access to our work by a non-expert community. Since the changes are extensive for this point and some of the others, the changes to the manuscript text are not repeated here, but rather in a copy of the manuscript they are indicated with highlights for checking.

Since the Auger decay is involving 2 delocalized valence electrons, I have troubles to consider the electronic decay as atomic-like as claimed in the second paragraph of the introduction.

This comment of the reviewer arises from the phrasing in our original manuscript. In the previous version of the manuscript, local decay channels in molecules were referred to as "Auger like". To avoid confusion, the term Auger decay is now only used to describe the isolated atomic case in the introduction. In the molecular case, the channels are only distinguished as local and non-local, as defined now in the introduction to the manuscript.

For me, the fourth paragraph is difficult to catch. The authors need to find a way to describe briefly the "macroscopic polarization". Without reading the reference 16, it is impossible to get an idea about this interesting concept.

The description of the ATAS experiment can be challenging. In order to better explain it, we have extended the description from one paragraph to two and added a figure showing the main features of the experiment. This added material allows us to go through the experiment in a stepwise fashion to guide the reader.

Concerning the results in figure 1, it would be interesting to know the experimental resolution, How an absorption spectrum measured with Synchrotron Radiation would look like ? Some signal seem to increase at 54.5 eV. Is it background ? Is it a discrete state ? Same question for the signal at 58.5 eV. I do not know the DELTA mod scale. I did not find explanation about this scale in the manuscript. How is it related to the delay ?

Photoelectron spectra of ICI have been collected at synchrotron facilities in the past and the associated publication is cited in the manuscript (Ref: J. Chem. Phys. 97, 7932 (1992)). As stated in the original manuscript, our spectral resolution is < 50 meV which competes favorably with the resolution available at synchrotron facilities. Collecting an absorption spectrum at a synchrotron will not yield complementary information to the one we present as the core-hole lifetimes become the limiting factor of the resolution.

The signal at 54.5 eV in the static spectrum originates from other core-excited states accessible to the 4d electron in this range. These excited states have lower transition dipole moments as they arise from dipole forbidden transitions (e.g. $4d^1 6s$ state). These states did not yield sufficiently detectable signals in the ATAS measurements and are therefore not discussed.

Similarly, the increase of absorption at 58.5 eV is due to the increasing contribution of direct ionization of the core-levels. The signal in ATAS relies on resonances, so these core-ionized molecules do not yield significant signals in ATAS.

The description of the acronym Δ mOD was added to the manuscript.

In the figure 2, the absorption spectrum is now upside down. Why ? For clarity, I advise to not change the orientation from figure 1 to figure 2. Again this DELTA mod scale on the y axis.

The spectrum in Fig 2 is not the upside down absorption spectrum, but the transient absorption spectrum at time zero as explained in the text. In order to better describe the figure and avoid further confusion, the text was modified, and the units of the graphic are explicitly labeled.

Page 7, I did not understand how the lifetime are determined. Is it a Voigt deconvolution ?

Lifetimes are obtained from the time evolution of negative features in the ATAS spectrum. As described in the text, lineouts are taken at the resonance energies and fitted to exponential decays (convoluted with a Gaussian for the instrument response at time zero). The time constants obtained yield the core-excited state lifetimes after application of a dividing factor of two. Each pair of features has

overlapping contributions and this is discussed in the SM in order to obtain the core-hole lifetimes.

I believe the most important questions concerns the understanding of the IR transition, the spectator Auger decay and the ICD-ETMD processes. It is very difficult to realize the details of the transitions, how do they evolve with the delay ?

The role of the NIR transition is discussed in detail in the text and in Figure 2a. The text lists the possible candidates for the interaction, i.e. resonant coupling, ionization of the excited molecules, and non-resonant AC stark shift. The two first candidates are disregarded when considering the energy diagram of ICI and the peak power density used in the experiment and the non-resonant AC Stark shift is indicated as the main possible candidate. To confirm that the non-resonant AC Stark shift dominates the interaction, the time zero spectrum is simulated by solving the time-dependent Schrödinger Equation using the experimental laser parameters and compared to the experimental results in Fig 2a. Similar results were found by a separate research group that did a similar experiment on iodomethane (J. Phys. Chem. Lett. 2019, 10, 2, 265–269) but where no lifetimes were reported.

The reviewer raises the question of the importance of participator channels in the core-excited state decay. Previous studies of related systems such as Xenon (Phys. Rev. A 1983, 28 (1), 261) or iodine atoms (Phys. Rev. A. 1992, 45, 2887) concluded that the participator channel is minor compared to the spectator channel. This is due to a weaker Coulombic interaction between the electron in the Rydberg state (here 6p) and the core-excited atom compared to the interaction between the valence electron. A similar argument will hold in the case of ICI and allows us to consider the spectator channels as contributing the most to the core-hole decay.

Concerning the role of ETMD and ICD in the core-hole decay, due to the delocalized nature of the valence orbital we cannot (and do not) use this terminology. Indeed, these processes are relevant in the decay of weakly-interacting atoms or molecules, where one can distinguish between the electrons of the different species. Here most of the valence orbitals are delocalized over both atoms and therefore despite the clear analogy we prefer to use the term non-local decay when referring to these processes.

Reviewer #2 (Remarks to the Author):

In the work “Attosecond spectroscopy reveals alignment dependent core-hole dynamics in the ICI molecule” by Marroux et al., the authors present the results of experiments using an isolated attosecond pulse (IAP) and a few-cycle infrared pulse to time-resolve core-excited state

dynamics in iodine monochloride molecules. Through measurements based on attosecond transient absorption spectroscopy (ATAS) they directly access the core-hole lifetimes with few-femtosecond resolution, demonstrating a dependence on the core-hole character, on timescales where the positions of the nuclei are essentially frozen. The experimental results are compared to thorough theoretical investigations, supporting the conclusion that the lifetimes are strongly affected by ligand-field effects, and thus on the alignment of the hole orbitals relative to the molecular axis.

The work is, to the best of my judgement, scientifically and technically sound and the combination of experimental and theoretical results strongly support the conclusions. The approach is novel in the sense that it is the first demonstration of ATAS of a weak-field process in core-excited molecules. I consider the work to be an important contribution to scientists in the field, in that it clearly demonstrates the precision and potential of ATAS measurements, and for the wider audience it opens up for few-femtosecond studies using ATAS in systems of interest in a range of science fields.

To that end, I recommend publication of the submitted manuscript in Nature Communications, but recommend the authors to consider the comments below in order to improve the accessibility and clarity of the manuscript:

1. Reading through the manuscript text, it is crucial to the understanding to keep track of the four different channels that the study is focusing on. I think that the notation and labelling of these between the different figures and between the text and the figures can be improved. For example, in Figure 1 they are referred to by their spectroscopic notation, in Figure 2 they are numbered 1-4 (and blue/red for the alignment character), in table 1 again with spectroscopic notation but no numbers and, finally, in the figures and table of the SM, they are sometimes referred to only by their energies. I encourage the authors to think about whether the clarity of this can be improved to facilitate for the reader. Maybe something like $1/2_{\{par/perp\}}$ or similar?

We thank the reviewer for his/her useful remarks and positive comments on our work. We have unified the notation throughout the manuscript in order to ease the readability of the text. As noted above, since the changes are extensive, the text changes are not repeated here, but a highlighted copy is included for checking.

2. I think that Figure 1 would benefit from some additional information. First, by adding axis labels in addition to the units. Second, Figure 1a is the same as Figure S2c (in the SM), and it would make sense, in the figure caption, to also describe the contents of Figure 1a) similarly to how it is done for Figure S2c (e.g. what is the red, blue, dashed, solid lines?).

We improved the overall didactic aspect of our manuscript and included the changes suggested by the reviewer in Fig 1. An extra panel was added to Fig 1 to present the main points of the experiment separately in order to improve the flow of the paper.

3. I find it slightly difficult to follow the discussion around the different times involved. The

fitted times (directly from the kinetic traces), the dephasing times (after subtracting the overlapping traces) and, finally, the lifetimes (after division by the factor of two due to the difference between the decay of the macroscopic polarization and the population lifetime). I think that it could be beneficial to present all of this in a single table in the SM, including all the times in a step-wise fashion, i.e. merging tables S2 and S3 (maybe even table S1 with the results from the lineshapes?) and adding the dephasing time as an intermediate step. In addition, I find that the reason for the factor of two is explained in the main text, with reference to the appropriate source, while in the SM it is (as far as I can find) only referred to as “Accounting for the factor of two...”. I think it would be useful to repeat the explanation and provide the reference again in the SM.

The results are separated into two tables in the supplementary material (S2 and S3) in order to separate the time constant obtained from the direct fit of the data, which includes overlapping contributions (S2), and the one where the contributions are separated, and lifetimes are then obtained. In order to guide the reader through the subtraction procedure, the last table was modified and now explicitly shows the dephasing times along with their source (i.e. direct fit vs. deconvolution).

Regarding the discussion of the relation between dephasing times and lifetimes in the SM, the description and references already present in the main text have been transposed to the paragraphs in the SM.

Reviewer #3 (Remarks to the Author):

This manuscript describes a beautiful study of core-hole decay dynamics in ICl using attosecond transient absorption spectroscopy (ATAS). The authors measure the lifetimes of $4d^{-1}6p$ Rydberg states of ICl and find lifetimes ranging from 3.5 to 6.9 fs. They convincingly argue that these decays are purely electronic with structural dynamics playing a negligible role. The data is convincing, the paper is clearly and nicely written and the overall interpretation is plausible. The main issue I have is the fact that spin-orbit coupling is neglected everywhere, both in the qualitative interpretation and in the calculations. I have some doubts that this is correct and it might have implications on the interpretation.

We thank the reviewer for his/her high appreciation of our work. We agree that the spin-orbit effects are important and play a role in the energetics. They also might affect the strength of the effect we discuss and, as we mention in the text, could be one of the reasons that our calculations predict shorter lifetimes. However, accounting for the spin-orbit effects should not change the main conclusion of the study, namely the different decay efficiency of holes along or perpendicular to the internuclear axis.

As evidence, we can compare the linewidth of two different spin states in a system where no ligand splitting is present. Xenon is a good candidate because the $4d$ levels have a spin-orbit splitting close to the one of iodine (1.9 eV for xenon and 1.75 eV in iodine) and many references exist for this system. (J. Phys.

B: At. Mol. Phys. 10 2479 (1977), J. Chem. Phys. 97, 7932 (1992) and J. Phys. B: At. Mol. Opt. Phys. 28 4529(1995)).

These studies measured a similar linewidth for core-holes in the $4d_{3/2}$ or $4d_{5/2}$ (i.e. 129 ± 8 meV) indicating that the spin state of the state considered does not affect significantly the decay. This indicates that the main effect on the core-hole decay is due to the alignment of the core-hole wave function with respect to the neighboring atom. Orbitals computed with the spin-orbit interaction show a strong variation in spatial distribution. This is evident in their decomposition reported in SM3 where, for example, core holes with $\Omega_c=3/2$ and $5/2$ characters are mainly composed of d_{xy} and $d_{x^2-y^2}$ orbitals. The angular momentum projections of these orbitals are high ($|L_z|=2$) indicating an electronic distribution perpendicular to the molecular axis. A detailed explanation was added to the manuscript in order to guide the reader.

First, the pictorial representation of the d orbitals as d_{z^2} , $d_{x^2-y^2}$, etc. completely neglects spin-orbital coupling. By analogy with p orbitals, it is clear that the spin-orbit-coupled eigenfunctions are actually linear combinations of the pure angular-momentum eigenfunctions, which are currently shown. This will change the qualitative shape of the orbitals, even in an atom. Since this is very easy to check, I suggest that the authors show the spin-orbit-coupled eigenstates, instead.

We performed SCF calculations using an infinite-order two-component relativistic Hamiltonian using the DIRAC program package and extracted the density of the corresponding 4d orbitals. Those are shown in SM3 . As indicated, their shape is only slightly different than in the non-relativistic case.

Second, and this is related, the authors use a Hund's case (a) notation throughout the paper, where a Hund's case (c) notation would be needed. Because the spin-orbit splitting is completely dominant here, λ cannot be a good quantum number, only ω can be. As a consequence, also S (which should be Σ in Hund's case (a)) will not be a good quantum number. I suggest that the authors change the quantum-number notation to be consistent with Hund's case (c), or explain their choice of the current notation.

We completely agree with the referee that it is more correct to use Hund's case (c) notation. We have changed the notation throughout the paper and added the decomposition of the resulting orbital into spin-orbital free orbital in SM3.

Third, and this might well be too demanding, it would be great to see some theoretical results including spin-orbit coupling.

Unfortunately, we do not possess a relativistic Fano-ADC-Stieltjes code and thus we are not able to perform such calculations. However, as already noted above, apart from some quantitative improvement of the decay-width values, we do not expect a qualitative difference. Our calculations already reproduce rather well the overall trend and the ratios between the decay widths.

Finally, I would like to suggest that the authors also reference previous work done on studying purely electronic dynamics in molecules using high-harmonic spectroscopy and RABBIT.

The needed references were added in the introduction of the paper.

REVIEWERS' COMMENTS

Reviewer #1 (Remarks to the Author):

I consider that the authors well responded to my questions and the manuscript is now pleasant to read. I recommend the publication of the manuscript as is.

Reviewer #2 (Remarks to the Author):

I find that the authors have significantly improved the accessibility and clarity of the manuscript following the initial review, and thus still recommend publication in Nature Communications.

Reviewer #3 (Remarks to the Author):

The authors have revised their manuscript very carefully and have successfully resolved most of the concerns of all 3 referees. I therefore recommend publication, subject to very minor revisions:

1) p. 6: the notation " $m=2$ " would more correctly read " $\lambda = 2$ "

2) the distinction between results including or excluding spin-orbit couplings is not always clear. In the main text, this could be added at the beginning of the section discussing the theory results and in Fig. 4. In the supplementary material, this could be stated at the beginning of SM4.

3) On p. 8, the " Δ " in " Δ^2 " is missing.

Overall, I congratulate the authors on an excellent paper and a valuable contribution to the literature.